# ESTIMATING TREATMENT EFFECTS VIA ORTHOGONAL REGULARIZATION

## ABSTRACT

Decision-making often requires accurate estimation of causal effects from observational data. This is challenging as outcomes of alternative decisions are not observed and have to be estimated. Previous methods estimate outcomes based on unconfoundedness but neglect any constraints that unconfoundedness imposes on the outcomes. In this paper, we propose a novel regularization framework in which we formalize unconfoundedness as an orthogonality constraint. We provide theoretical guarantees that this yields an asymptotically normal estimator for the average causal effect. Compared to other estimators, its asymptotic variance is strictly smaller. Based on our regularization framework, we develop **d**eep **o**rthogonal **n**etworks for **u**nconfounded **t**reatments (DONUT) which learn outcomes that are orthogonal to the treatment assignment. Using a variety of benchmark datasets for causal inference, we demonstrate that DONUT outperforms the state-of-the-art substantially.

## 1 INTRODUCTION

Estimating the causal effect of an intervention (i. e., treatment effect) is integral for individual decision making in many domains such as marketing (Brodersen et al., 2015; Hatt & Feuerriegel, 2020), economics (Heckman et al., 1997), and epidemiology (Robins et al., 2000). For instance, in order to control an epidemic, it is relevant for public decision-makers to estimate the causal effect of school-closures (intervention) on the infection rate (outcome).

The causal effect of an intervention can be estimated in two ways: *randomized control trials (RCTs)* and *observational studies*. RCTs are widely recognized as the gold standard for estimating causal effects, yet conducting RCTs is often infeasible (Robins et al., 2000). For instance, randomly allocating different policy interventions during an epidemic might be unethical and impractical. Unlike RCTs, observational studies adopt observed data to infer causal effects. For this, covariates must be collected that contain all confounders (i. e., variables that affect both treatment and outcome). This is becoming increasingly common due to ease of access to rich data. In this paper, we estimate the *average causal effect* of a treatment from observational data.

In order to estimate the causal effect of a treatment, the outcome of an alternative treatment has to be estimated. However, this is challenging, since we do not know what the outcome would have been if another treatment had been applied. Existing methods for estimating treatment effects use the treatment assignment as a feature and train regression models to estimate the outcomes (Funk et al., 2011; Kallus, 2017b). Methods based on nearest neighbors and matching are adopted to find similar subjects (Ho et al., 2007; Crump et al., 2008; Kallus, 2017a; 2020). Tree and forest-based methods (Wager & Athey, 2018) estimate the treatment effect at the leaf node and train many weak learners to build expressive ensemble models. Gaussian process-based methods provide uncertainty quantification (Alaa & van der Schaar, 2017; Ray & Szabo, 2019). Weighting-based approaches re-weight the outcomes using weights based on covariate and treatment data (Kallus, 2018). For instance, Fong et al. (2018); Yiu & Su (2018) seek weights such that the treatment assignment is unassociated with the covariates. However, they do not require the treatment assignment to be unassociated with the potential outcomes. Doubly robust methods combine a model for the outcomes and a model for the treatment propensity in a manner that is robust to misspecification (Funk et al., 2011; Benkeser et al., 2017; Chernozhukov et al., 2018). Recently, deep learning has been successful for this task due to its strong predictive performance and ability to learn representations of the data

(e. g., Johansson et al., 2016; Louizos et al., 2017; Shalit et al., 2017; Yao et al., 2018; Yoon et al., 2018; Shi et al., 2019). To ensure identifiability of the causal effect, state-of-the-art methods for estimating treatment effects are based on unconfoundedness (i. e., all confounders are measured, and thus included in the covariates). Hence, unconfoundedness is assumed for identifiability, yet, during estimation of the model parameters, any implications on the unobserved outcomes that arise from unconfoundedness have been neglected.

**Contribution.**[1] In this paper, (i) we introduce a regularization framework that exploits unconfoundedness . To this end, we formalize unconfoundedness as an orthogonality constraint. This constraint is used during estimation of the model parameters to ensure that the outcomes are orthogonal to the treatment assignment. We prove sufficient conditions under which this yields an asymptotically normal estimator for the average causal effect. Compared to other estimators, its asymptotic variance is strictly smaller. (ii) Based on our regularization framework, we develop **d**eep **o**rthogonal **n**etworks for **u**nconfounded **t**reatments (**DONUT**) for estimating average causal effects. DONUT leverages the predictive capabilities of neural networks to learn outcomes that are orthogonal to the treatment assignment. Using a variety of benchmark datasets for causal inference, we demonstrate that DONUT outperforms the state-of-the-art substantially.

## 2 PROBLEM SETUP

Our objective is to estimate the average treatment effect (ATE) of a binary treatment from observational data. For this, we build upon the Neyman-Rubin potential outcomes framework (Rubin, 2005). Consider a population where every subject $i$ is described by the $d$-dimensional covariates $X_i \in \mathcal{R}^d$. Each subject is assigned a treatment $T_i \in \{0, 1\}$. The random variable $Y_i(1)$ corresponds to the outcome under treatment, i. e., $T_i = 1$, whereas $Y_i(0)$ corresponds to the outcome under *no* treatment, i. e., $T_i = 0$. These two random variables, $Y_i(1), Y_i(0) \in \mathbb{R}$, are known as the potential outcomes. Due to the fundamental problem of causal inference, only one of the potential outcomes is observed, but never both. The observed outcome is denoted by $Y_i$.

Our aim is to estimate the *average treatment effect*

$$\psi = \mathbb{E}[Y(1) - Y(0)]. \tag{1}$$

The following standard assumptions are sufficient for identifiability of the causal effect (Imbens & Rubin, 2015): *consistency* (i. e., $\forall t \in \{0, 1\} : Y = Y(t)$, if $T = t$); *positivity* (i. e., $\forall x \in \mathcal{R}^d : 0 < \mathbb{P}(T = 1 \mid X = x) < 1$); and *unconfoundedness*. Unconfoundedness assumes that all confounders are measured and, hence, conditioning on them blocks all backdoor paths. This is equivalent to assuming that the potential outcomes $Y(1)$ and $Y(0)$ are independent of the assigned treatment $T$ given the covariates $X$, i. e.,

$$Y(1), Y(0) \perp\!\!\!\perp T \mid X. \tag{2}$$

Based on this, the ATE is equal to

$$\psi = \mathbb{E}[\mathbb{E}[Y \mid X, T = 1] - \mathbb{E}[Y \mid X, T = 0]]. \tag{3}$$

Our task is to estimate the function $f(x, t) = \mathbb{E}[Y \mid X = x, T = t]$ for all $x \in \mathcal{R}^d$ and $t \in \{0, 1\}$ based on observational data $\mathcal{D} = \{(X_i, T_i, Y_i)\}_{i=1}^n$.

## 3 ORTHOGONAL REGULARIZATION FOR ESTIMATING TREATMENT EFFECTS

The key idea of our regularization framework is to exploit the implications on the outcomes, that result from unconfoundedness. For this, we formalize unconfoundedness as an orthogonality constraint. This orthogonality constraint ensures that the outcomes are orthogonal to the treatment assignment. We later introduce a specific variant of our regularization framework based on neural networks, which yields DONUT.

---

[1]Code available at `github.com/anonymous/donut` (anonymized for peer-review).

### 3.1 UNCONFOUNDEDNESS AS ORTHOGONALITY CONSTRAINT

Under unconfoundedness, the outcomes are independent of the assigned treatment given the covariates, i. e.,

$$Y(1), Y(0) \perp\!\!\!\perp T \mid X. \tag{4}$$

Using the inner product $\langle V, W \rangle = \frac{1}{n} \sum_{i=1}^{n} V_i W_i$, we formalize the following *orthogonality constraint*:

$$\langle Y(t) - f(X, t), T - \pi(X) \rangle = \frac{1}{n} \sum_{i=1}^{n} (Y_i(t) - f(X_i, t))(T_i - \pi(X_i)) = 0, \tag{5}$$

for $t \in \{0, 1\}$, and where $f(x, t) = \mathbb{E}[Y \mid X = x, T = t]$ and $\pi(x) = \mathbb{E}[T \mid X = x]$. The function $\pi$ is the *propensity score*.[2] This is a necessary condition for unconfoundedness, since unconfoundedness requires the covariance between $Y(t)$ and $T$ given $X$ to be zero due to the independence of the outcomes and the treatment assignment. Hence, since the inner product in (5) is the empirical covariance between $Y(t)$ and $T$ given $X$, unconfoundedness requires the (centered) outcomes to be orthogonal to the (centered) treatment assignment with respect to the above inner product.[3] An appropriate method for estimating outcomes based on unconfoundedness should ensure that the orthogonality constraint holds. Although existing methods are based on unconfoundedness, the orthogonality of the outcomes to the treatment assignment is ignored during estimation of the model parameters. As a remedy, we propose a regularization framework that accommodates the orthogonality constraint in the estimation procedure.

### 3.2 PROPOSED REGULARIZATION FRAMEWORK

In our regularization framework, the orthogonality constraint in (5) is included in the estimation procedure as follows. Let $\mathcal{H}_f \subseteq [\mathcal{R}^d \times \{0, 1\} \to \mathbb{R}]$ and $\mathcal{H}_\pi \subseteq [\mathcal{R}^d \to [0, 1]]$ be function classes for $f$ and $\pi$, and $\epsilon \in \mathbb{R}$ a model parameter. Then, the objective is to find the solution to the optimization problem

$$\inf_{(f, \pi, \epsilon) \in \mathcal{H}_f \times \mathcal{H}_\pi \times \mathbb{R}} \mathcal{L}_{FL}(f, \pi; \mathcal{D}) + \lambda \, \Omega_{OR}(f, \pi, \epsilon; \mathcal{D}), \tag{6}$$

where $\mathcal{L}_{FL}$ is the so-called factual loss (see Section 3.2.1) between the estimated models and observed (factual) data. The term $\Omega_{OR}$ is our new orthogonal regularization that enforces the orthogonality constraint using the model parameter $\epsilon$ (see Section 3.2.2). The variable $\lambda \in \mathbb{R}_+$ is a hyperparameter controlling the strength of regularization. We describe both components of (6), i. e., factual loss and orthogonal regularization, in the following.

### 3.2.1 FACTUAL LOSS $\mathcal{L}_{FL}$

Similar to previous work (e. g., Shi et al., 2019), the outcome model $f$ and propensity score model $\pi$ are estimated using the observed data and the factual loss given by

$$\mathcal{L}_{FL}(f, \pi; \mathcal{D}) = \frac{1}{n} \sum_{i=1}^{n} (f(X_i, T_i) - Y_i)^2 + \alpha \, \mathrm{CrossEntropy}(\pi(X_i), T_i), \tag{7}$$

where $\alpha \in \mathbb{R}_+$ is a hyperparameter weighting the terms of the factual loss. We note the following about the estimation of the outcome model $f$ in (7). In the first term of the factual loss, $f$ is fitted to the observed outcomes and, thus, no information about the unobserved outcomes is used. As a consequence, the model learns about the observed outcomes, but not about the unobserved outcomes. State-of-the-art methods *exclusively* rely on the factual loss to estimate the outcomes (e. g. Shalit et al., 2017). This is based on the assumptions that similar subjects have similar outcomes (Yao et al., 2018) and that for every subject in the data, there is a similar subject in the data with the opposite treatment. However, this is unlikely in presence of selection bias, i. e., when treatment and control groups differ systemically. This pitfall is addressed by our orthogonal regularization in the following.

---

[2]Note that the propensity score is defined as $\pi(x) = \mathbb{P}(T = 1 \mid X = x)$ (Rosenbaum & Rubin, 1983), which is equivalent to $\mathbb{E}[T \mid X = x]$ for binary treatments.

[3]Note that we use the notion orthogonality with respect to the inner product in (5). This is different from the notion of Neyman orthogonality in (Nie & Wager, 2017; Chernozhukov et al., 2018), which requires the Gâteau derivative of a debiased score function to vanish at the true parameter. In contrast, our orthogonality constraint exploits unconfoundedness to ensure that the outcomes are orthogonal (w.r.t. the inner product in (5)) to the treatment assignment.

### 3.2.2 Orthogonal Regularization $\Omega_{OR}$

We now specify the orthogonal regularization term $\Omega_{OR}$ that ensures that the orthogonality constraint in (5) is satisfied. We first state $\Omega_{OR}$ and then explain each of its parts. The orthogonal regularization term is given by

$$\Omega_{OR}(f, \pi, \epsilon; \mathcal{D}) = \frac{1}{n} \sum_{i=1}^{n} (Y_i^*(0) - f^\epsilon(X_i, 0))^2, \tag{8}$$

with

$$Y_i^*(0) = Y_i - \psi^* T_i, \quad \psi^* = \frac{1}{n} \sum_{i=1}^{n} f(X_i, 1) - f(X_i, 0) \tag{9}$$

and

$$f^\epsilon(X_i, t) = f(X_i, t) + \epsilon(T_i - \pi(X_i)), \text{ for } t \in \{0, 1\}, \tag{10}$$

where the pseudo outcome $Y_i^*(0)$ and the perturbation function $f^\epsilon$ are required to learn outcomes that are orthogonal to the treatment assignment, as described in the following.

**Pseudo outcome.** The untreated outcome of a subject can be expressed as $Y_i(0) = Y_i - \psi(X_i) T_i$, where $\psi(X) = f(X, 1) - f(X, 0)$ is the true treatment effect at $X$. Hence, if we had access to the treatment effect $\psi(X)$, we would also have access to the untreated outcome $Y_i(0)$ even if we did not observe it.[4] We use the average treatment effect of the current model fit, i. e., $\psi^*$ in (9), as proxy for $\psi(X_i)$. This creates a pseudo outcome, $Y_i^*(0) = Y_i - \psi^* T_i$. Under sufficient conditions, this converges to the true outcome (see Section 4).

**Perturbation function.** In order to ensure that the orthogonality constraint is satisfied, i. e., $\langle Y(t) - f(X, t), T - \pi(X) \rangle = 0$, we use a perturbation function $f^\epsilon$. Simply adding the inner product as a regularization term is not sufficient, since this does *not* guarantee that the orthogonality constraint holds at a solution of (6). A similar approach is used in targeted minimum loss estimation (e. g., van der Laan & Rose, 2011) and, more general, dual problems in optimization (Zalinescu, 2002). We extend the function $f$ to a perturbation function $f^\epsilon$ using the model parameter $\epsilon$ as in (10). As a result, solving the optimization problem in (6) forces the outcome estimates $f^\epsilon$ to satisfy the orthogonality constraint. Mathematically, this can be seen by taking the partial derivative of (6) and setting it to zero, i. e.,

$$0 = \frac{\partial}{\partial \epsilon} (\mathcal{L}_{FL} + \lambda \, \Omega_{OR}(f, \pi, \epsilon; \mathcal{D})) \Big|_{\epsilon = \hat{\epsilon}} = 2 \frac{1}{n} \sum_{i=1}^{n} (Y_i^*(0) - f^\epsilon(X_i, 0)) \, (T_i - \pi(X_i)), \tag{11}$$

and, hence, $\langle Y^*(0) - f^\epsilon(X, 0), T - \pi(X) \rangle = 0$. It is analytically sufficient to regularize only the untreated outcome.[5] Whenever one of the outcomes is orthogonal to the treatment assignment, the other outcome is also orthogonal. A proof of this is provided in Appendix A.

Then, the outcomes are estimated using $f^\epsilon(x, t)$ and, therefore, the estimate for the ATE is $\hat{\psi} = \frac{1}{n} \sum_{i=1}^{n} \hat{f}^\epsilon(X_i, 1) - \hat{f}^\epsilon(X_i, 0)$, which coincides with $\psi^*$, since the perturbation terms in (10) cancel.

### 3.3 Deep Orthogonal Networks for Unconfounded Treatments

Our regularization framework works with any outcome model $f$ and propensity score model $\pi$ that is estimated through a loss function. The regularization framework ensures that the outcomes are estimated such that they are orthogonal to the treatment assignment. We introduce a specific variant of our regularization framework based on feedforward neural networks, which yields **d**eep **o**rthogonal **n**etworks for **u**nconfounded **t**reatments (**DONUT**). Neural networks present a suitable model class due to their strong predictive performance. The architecture of DONUT is set as follows. For the outcome model $f$, we use the basic architecture of TARNet (Shalit et al., 2017). TARNet uses a deep feedforward neural network to produce a representation layer, followed by two 2-layer neural networks to predict each of the potential outcomes from the shared representation. For the propensity

---

[4] This can be seen by distinction of cases: if $T_i = 1$, then $Y_i(0) = Y_i - \psi(X_i)$, and if $T_i = 0$, then $Y_i(0) = Y_i$.

[5] It is straightforward to extend the orthogonal regularization term to incorporate both outcomes by adding the same term for the treated outcome $Y(1)$, including the pseudo outcome $Y^*(1)$ and the perturbation function $f^\epsilon(x, 1)$ to enforce orthogonality. We demonstrate this in Appendix A.

score model $\pi$, we use a logistic regression (Rosenbaum & Rubin, 1983). The choice of feedforward neural networks is particularly strengthened by the theoretical discussion in Section 4.

## 4 THEORETICAL GUARANTEES

We prove sufficient conditions under which our regularization framework yields an asymptotically normal[6] estimator $\hat{\psi}$ for the true ATE $\psi$. Asymptotic normality is particularly favorable for an estimator, as such estimators converge to the true ATE at a rate $1/\sqrt{n}$. We further show that, compared to other estimators, our estimator has strictly smaller asymptotic variance.

**Theorem 1.** *(Asymptotic Normality.) Suppose*

1. *The estimator $\hat{\eta} = (\hat{f}, \hat{\pi})$ for the outcome and propensity score model converges to some $\bar{\eta} = (\bar{f}, \bar{\pi})$ in the sense that $\|\hat{\eta} - \bar{\eta}\| = o_p(1)$, where either $\bar{f} = f$ or $\bar{\pi} = \pi$ (or both) corresponds to the true function.*

2. *The treatment effect is homogeneous, i. e., $\forall x \in \mathcal{R}^d : \psi(x) = \psi$.*

3. *The estimators $\hat{f}$ and $\hat{\pi}$ take values in $\mathbb{P}$-Donsker classes, i. e., $\mathcal{H}_f, \mathcal{H}_\pi \in CLT(\mathbb{P})$.[7]*

*Then $\hat{\psi}$ is asymptotically normal, i. e.,*

$$\sqrt{n}\,(\hat{\psi} - \psi) \xrightarrow{d} \mathcal{N}\Big(0, \frac{\sigma^2}{\mathbb{E}[\mathrm{Var}(T \mid X)]}\Big), \tag{12}$$

*where $\sigma^2$ is the variance of the outcome.* □

The proof is provided in Appendix C. We make the following remarks about the result and its conditions. Condition 1 requires that either the model for the outcomes $f$ or the model for the propensity score $\pi$ (or both) is correctly specified, but not necessarily both. This means that our estimator is *doubly robust*, since it is consistent under correct specification of either $f$ or $\pi$. In particular, neural networks converge at a fast enough rate to invoke the doubly robustness condition (Farrell et al., 2018). Condition 2 can be easily relaxed to any specification of $\psi$ as long as it has finitely many parameters and given that the appropriate identification criteria hold. This changes the orthogonal regularization in (8) accordingly; see Appendix D for details. However, the advantage of Condition 2 is that it explains why the asymptotic variance of $\hat{\psi}$ is smaller compared to other estimators. In particular, the difference in asymptotic variance to some estimators (e. g., the inverse probability weighted estimator) can be sizable if the propensity score is close to zero or one. Often, this difference is not offset by weaker restrictions imposed by heterogeneous treatment effects (e. g., Vansteelandt & Joffe, 2014). Condition 3 captures a large class of functions of estimators for $f$ and $\pi$ from which we can choose. An overview of $\mathbb{P}$-Donsker classes is given in Mikosch et al. (1997). In particular, the class of feedforward neural networks is a $\mathbb{P}$-Donsker class, since Lipschitz parametric functions are $\mathbb{P}$-Donsker functions, and any Lipschitz transformation of a $\mathbb{P}$-Donsker function is again a $\mathbb{P}$-Donsker function. For mathematical details, see Appendix E. Together with the convergence rate for doubly robustness, this provides theoretical justification for the use of neural networks in our regularization framework, and therefore in DONUT.

**Similarity to partially linear regression.** We find an interesting similarity between our estimator $\hat{\psi}$ and the estimator obtained by partially linear regression (PLR) (see (1.5) in Chernozhukov et al. (2018)). The analytical expression of $\hat{\psi}$ yield by our regularization framework in (8) is given in (31) in Appendix C. This is similar to the estimator obtained by PLR after partialling the effect of $X$ out from $T$ (see (1.5) in Chernozhukov et al. (2018)). However, the PLR separately estimates the nuisance functions $\hat{f}$ and $\hat{\pi}$ and then plugs them into the estimator. Our approach does not use the analytical expression as plug-in estimator, but the estimator arises directly from solving the

---

[6]We refer to Appendix B for a brief discussion on asymptotically normal estimators.

[7]A class $\mathcal{F}$ of measurable functions on a probability space $(\Omega, \mathcal{A}, \mathbb{P})$ is called a $\mathbb{P}$-Donsker class if, for $\mathbb{G}_n^{\mathbb{P}} = \sqrt{n}(\mathbb{P}_n - \mathbb{P})$, the empirical process $\{\mathbb{G}_n^{\mathbb{P}} f : f \in \mathcal{F}\}_{n \geq 1}$ converges weakly to a $\mathbb{P}$-Brownian bridge $\mathbb{G}^{\mathbb{P}}$. Donsker classes include parametric classes, but also many other classes, including infinite-dimensional classes, e. g., smooth functions and bounded monotone functions. See (Mikosch et al., 1997) for more details.

optimization problem in (6) via (8). The difference becomes apparent in the experiments (Section 5), where we include the PLR as a baseline and find that the performance of DONUT is superior.

**Comparison to other estimators.** We compare the asymptotic behavior of our estimator $\hat{\psi}$ to other regular estimators (e. g., Funk et al. (2011); Nie & Wager (2017)). We find that, under the conditions of Theorem 1, there does not exist a regular estimator that achieves strictly smaller asymptotic variance than our estimator. The reason for this is as follows. We proof in Appendix C that our regularization framework yields an asymptotically normal estimator for the ATE. In particular, we proof that our estimator is efficient (see Appendix C for the efficient influence function), and therefore it achieves the efficiency bound. As a consequence, among all regular estimators, our estimator achieves the smallest asymptotic variance. In Appendix G, we make an example in which we compare the inverse probability weighted estimator to our estimator. We show that the difference in asymptotic variance becomes particularly pronounced in presence of selection bias (i. e., when treatment and control groups differ systemically).

To summarize, we prove sufficient conditions under which our estimator $\hat{\psi}$ is asymptotically normal. Hence, our proposed estimator converges to the true ATE $\psi$ at a fast rate. Compared to other regular estimators, its asymptotic variance is strictly smaller, since we prove that our estimator is efficient. Since feedforward neural networks converge at a fast enough rate to invoke doubly robustness and belong to a $\mathbb{P}$-Donsker class, these results hold true when using feedforward neural networks in our regularization framework, which yields DONUT.

## 5 EXPERIMENTS

Our proposed DONUT is evaluated against state-of-the-art baselines, where we find that its ATE estimation is superior (Section 5.2). Its benefits appear especially for problems subject to selection bias, which is confirmed as part of a simulation study (Section 5.3). In the latter, the theoretical guarantees from Section 4 are confirmed.

### 5.1 SETUP

Evaluating methods for estimating causal effects is challenging as we rarely have access to ground truth causal effects. Established procedures for evaluation of such methods rely on semi-synthetic data, which reflect the real world. Our experimental setup follows established procedure regarding datasets, baselines, and performance metrics (e. g., Johansson et al., 2016; Shalit et al., 2017).

**Datasets.** We evaluate all methods across four benchmark datasets for causal inference: **IHDP** (e. g., Johansson et al., 2016), **Twins** (e. g., Yoon et al., 2018), **ACIC 2018** (e. g., Shi et al., 2019), and **Jobs** (e. g., Shalit et al., 2017). The first three datasets are semi-synthetic, while the last originated from a RCT. Details on IHDP, Twins, ACIC 2018, and Jobs are provided in Appendix H.

**Training details.** DONUT is trained using the regularization framework in (6), where both the outcome model and the propensity score model are trained jointly using stochastic gradient descent with momentum. The hidden layer size is 200 for the representation layers and 100 for the outcome layers similar to Shalit et al. (2017); Shi et al. (2019). The hyperparameter $\alpha$ in the factual loss (7) is set to 1 and $\lambda$ in the orthogonal regularization (8) is determined by hyperparameter optimization over $\{10^k\}_{k=-2}^2$. For IHDP, we follow established practice (e. g., Shalit et al., 2017) and average over 1,000 realizations of the outcomes with 63/27/10 train/validation/test splits. Following Shalit et al. (2017); Yoon et al. (2018); Shi et al. (2019), we average over 100 different train/validation/test splits for Twins and Jobs, and over 10 splits for each dataset for ACIC, all with ratios 56/24/20.

**Baselines.** We compare DONUT against 17 state-of-the-art methods for estimating treatment effects, organized in the following groups: (i) Regression methods: Linear regression with treatment as covariate (**OLS/LR-1**), separate linear regressors for each treatment (**OLS/LR-2**), and balancing linear regression (**BLR**) (Johansson et al., 2016); (ii) Matching methods: $k$-nearest neighbor ($k$-**NN**) (Crump et al., 2008); (iii) Tree methods: Bayesian additive regression trees (**BART**) (Chipman et al., 2012), random forest (**R-Forest**) (Breiman, 2001), and causal forest (**C-Forest**) (Wager & Athey, 2018); (iv) Gaussian process methods: Causal multi-task Gaussian process (**CMGP**) (Alaa & van der Schaar, 2017) and debiased Gaussian process (**D-GP**) (Ray & Szabo, 2019); (v) Neural network methods: Balancing neural network (**BNN**) (Johansson et al., 2016), treatment-agnostic

Table 1: Results for estimating average treatment effects on IHDP, Twins, and Jobs. Lower is better.

**Results**

| | IHDP ($\epsilon_{\text{ATE}}$) | | TWINS ($\epsilon_{\text{ATE}}$) | | JOBS ($\epsilon_{\text{ATT}}$) | |
| --- | --- | --- | --- | --- | --- | --- |
| | | | Datasets (Mean $\pm$ Std) | | | |
| METHOD | IN-S. | OUT-S. | IN-S. | OUT-S. | IN-S. | OUT-S. |
| OLS/LR-1 | $.73 \pm .04$ | $.94 \pm .06$ | $.0038 \pm .0025$ | $.0069 \pm .0056$ | $.01 \pm .00$ | $.08 \pm .04$ |
| OLS/LR-2 | $.14 \pm .01$ | $.31 \pm .02$ | $.0039 \pm .0025$ | $.0070 \pm .0059$ | $.01 \pm .01$ | $.08 \pm .03$ |
| BLR | $.72 \pm .04$ | $.93 \pm .05$ | $.0057 \pm .0036$ | $.0334 \pm .0092$ | $.01 \pm .01$ | $.08 \pm .03$ |
| $k$-NN | $.14 \pm .01$ | $.79 \pm .05$ | $.0028 \pm .0021$ | $.0051 \pm .0039$ | $.21 \pm .01$ | $.13 \pm .05$ |
| BART | $.23 \pm .01$ | $.34 \pm .02$ | $.1206 \pm .0236$ | $.1265 \pm .0234$ | $.02 \pm .00$ | $.08 \pm .03$ |
| R-FOREST | $.73 \pm .05$ | $.96 \pm .06$ | $.0049 \pm .0034$ | $.0080 \pm .0051$ | $.03 \pm .01$ | $.09 \pm .04$ |
| C-FOREST | $.18 \pm .01$ | $.40 \pm .03$ | $.0286 \pm .0035$ | $.0335 \pm .0083$ | $.03 \pm .01$ | $.07 \pm .03$ |
| D-GP | $.14 \pm .32$ | $.17 \pm .47$ | $.0046 \pm .0033$ | $.0058 \pm .0043$ | $.03 \pm .01$ | $.06 \pm .05$ |
| CMGP | $\mathbf{.11 \pm .10}$ | $\mathbf{.13 \pm .12}$ | $.0124 \pm .0051$ | $.0143 \pm .0116$ | $.06 \pm .06$ | $.09 \pm .07$ |
| BNN | $.37 \pm .03$ | $.42 \pm .03$ | $.0056 \pm .0032$ | $.0203 \pm .0071$ | $.04 \pm .01$ | $.09 \pm .04$ |
| TARNET | $.26 \pm .01$ | $.28 \pm .01$ | $.0108 \pm .0017$ | $.0151 \pm .0018$ | $.05 \pm .02$ | $.11 \pm .04$ |
| CFR-WASS | $.25 \pm .01$ | $.27 \pm .01$ | $.0112 \pm .0016$ | $.0284 \pm .0032$ | $.04 \pm .01$ | $.09 \pm .03$ |
| GANITE | $.43 \pm .05$ | $.49 \pm .05$ | $.0058 \pm .0017$ | $.0089 \pm .0075$ | $.01 \pm .01$ | $.06 \pm .03$ |
| AIPWE | $.13 \pm .12$ | $.22 \pm .28$ | $.0027 \pm .0013$ | $.0048 \pm .0032$ | $.03 \pm .01$ | $.10 \pm .08$ |
| TMLE | $.13 \pm .10$ | $.33 \pm .32$ | $.0034 \pm .0020$ | $.0053 \pm .0025$ | $.02 \pm .01$ | $.06 \pm .05$ |
| PLR | $.63 \pm .09$ | $1.32 \pm .31$ | $.0133 \pm .0255$ | $.0084 \pm .0354$ | $.09 \pm .04$ | $.10 \pm .08$ |
| DRAGONNET | $.14 \pm .01$ | $.20 \pm .05$ | $.0062 \pm .0051$ | $.0064 \pm .0054$ | $.02 \pm .01$ | $.06 \pm .05$ |
| **DONUT** | $.13 \pm .01$ | $.19 \pm .02$ | $\mathbf{.0025 \pm .0016}$ | $\mathbf{.0033 \pm .0026}$ | $\mathbf{.01 \pm .00}$ | $\mathbf{.06 \pm .05}$ |

representation network (**TARNet**) (Shalit et al., 2017), counterfactual regression with Wasserstein distance (**CFR-WASS**) (Shalit et al., 2017), generative adversarial networks (**GANITE**) (Yoon et al., 2018), and **Dragonnet** (Shi et al., 2019); (vi) Plug-in estimators: Augmented inverse probability weighted estimator (**AIPWE**) (Cao et al., 2009), targeted maximum likelihood estimator (**TMLE**) (van der Laan & Rubin, 2006), and partially linear regression (**PLR**) (Chernozhukov et al., 2018) all of them using the outcome and propensity score model of Dragonnet as nuisance functions.

**Performance metrics.** Following established procedure, we report the following metrics for each dataset. For IHDP and ACIC 2018, we use the absolute error in average treatment effect (Johansson et al., 2016): $\epsilon_{\text{ATE}} = |\frac{1}{n} \sum_{i=1}^{n}(f(x_i, 1) - f(x_i, 0)) - \frac{1}{n} \sum_{i=1}^{n}(\hat{f}(x_i, 1) - \hat{f}(x_i, 0))|$. For Twins, we use the absolute error in observed average treatment effect (Yoon et al., 2018): $\epsilon_{\text{ATE}} = |\frac{1}{n} \sum_{i=1}^{n}(y_i(1) - y_i(0)) - \frac{1}{n} \sum_{i=1}^{n}(\hat{y}_i(1) - \hat{y}_i(0))|$. For Jobs, all treated subjects $\mathcal{T}$ were part of the original randomized sample $\mathcal{E}$, and hence, the true average treatment effect can be computed on the treated by ATT $= |\mathcal{T}|^{-1} \sum_{i \in \mathcal{T}} y_i - |\mathcal{C} \cap \mathcal{E}|^{-1} \sum_{i \in \mathcal{C} \cap \mathcal{E}} y_i$, where $\mathcal{C}$ is the control group. Similar to (Shalit et al., 2017), we then use the error: $\epsilon_{\text{ATT}} = |\text{ATT} - |\mathcal{T}|^{-1} \sum_{i \in \mathcal{T}}(\hat{f}(x_i, 1) - \hat{f}(x_i, 0))|$.

## 5.2 RESULTS

IHDP, Twins, and Jobs have been used to evaluate many methods for estimating treatment effects. In Table 1, the results of the experiments on IHDP, Twins, and Jobs are presented. Overall, DONUT achieves competitive performance across all datasets. On IHDP, the CMGP baseline achieves slightly lower estimation error (mean: .11 (CMGP) vs. .13 (DONUT)), but at the drawback of an inferior standard deviation (std.: .10 (CMGP) vs. .01 (DONUT)). However, as shown by previous work, the small sample size and the limited simulation settings of IHDP make it difficult to draw conclusions about methods (e. g., Yoon et al., 2018; Shi et al., 2019). In contrast, DONUT achieves superior performance on both Twins and Jobs, where the number of samples is much larger (Twins: $n =$11,400; Jobs: $n =$3,212). On these datasets, DONUT is state-of-the-art. In particular, we point out the difference to PLR which uses the analyitcal expression of our estimator similar to Chernozhukov et al. (2018) as discussed in Section 4, but as a plug-in estimator. For a reasonable comparison, we use the outcome models of Dragonnet as nuisance functions. We find that DONUT achieves superior performance.

Among neural network-based methods (i. e., BNN, TARNet, CFR-WASS, GANITE, Dragonnet, and DONUT), DONUT performs superior across all datasets. In comparison to TARNet, which shares the basic architecture of DONUT (but without orthogonal regularization), we achieve substantial reduction in ATE estimation error across

all datasets (out-of-sample error reduction: 32.1 % on IHDP, 78.1 % on Twins, and 45.5 % on Jobs). This demonstrates the effectiveness of our regularization framework.

We further evaluate DONUT using ACIC 2018. This collection of datasets was introduced for evaluating neural networks for estimating average treatment effects (Shi et al., 2019). We compare DONUT against the most competitive, and current state-of-the-art method, i. e., Dragonnet. In addition, we compare against TARNet (i. e., DONUT, but *without* orthogonal regularization). Table 2 presents the results of the experiments on ACIC 2018. The main observation is that DONUT improves estimation relative to TARNet (DONUT without orthogonal regularization) and relative to Dragonnet across a large collection of datasets. These results further confirm the benefit of including orthogonal regularization in the estimation procedure.

Table 2: Results for estimating average treatment effects on ACIC 2018. Comparison between Dragonnet (most competitive baseline), TARNet (same architecture as DONUT, but without orthogonal regularization) and DONUT. Lower is better.

**Results**

| METHOD | ACIC ($\epsilon_{\text{ATE}}$) | |
| --- | --- | --- |
| | IN-S. | OUT-S. |
| TARNET | $4.53 \pm 0.72$ | $4.48 \pm 0.74$ |
| DRAGONNET | $2.97 \pm 1.46$ | $2.99 \pm 1.48$ |
| **DONUT** | $\mathbf{1.22 \pm .33}$ | $\mathbf{1.26 \pm 0.29}$ |

### 5.3 SIMULATION STUDY ON SELECTION BIAS

To evaluate the robustness of DONUT with regards to selection bias (i. e., when treatment and control groups differ substantially), we generate synthetic data with varying selection bias according to a similar protocol as in Yao et al. (2018); Yoon et al. (2018). Details on the protocol are provided in Appendix I.

We compare DONUT against Dragonnet, which makes use of the AIPWE to estimate the average treatment effect. This is particular interesting for the comparison with DONUT due to the results in Section 4. We showed that, compared to any other estimator, our estimator obtains strictly smaller asymptotic variance, especially in presence of selection bias. Figure 1 presents the mean and standard deviation of $\epsilon_{\text{ATE}}$ for varying selection bias. We report two major insights: (i) Dragonnet is outperformed by DONUT across different levels of selection bias. As selection bias increases, the estimation error of DONUT remains stable. (ii) The standard deviation of both DONUT and Dragonnet increases as selection bias increases. However, in line with our findings that DONUT has the smallest asymptotic variance among all regular estimators, the standard deviation of DONUT remains consistently smaller than the standard deviation of Dragonnet. In addition, this difference becomes more pronounced the larger the selection bias. Hence, we observe empirical properties of our estimator that coincide with the theory derived in Section 4.

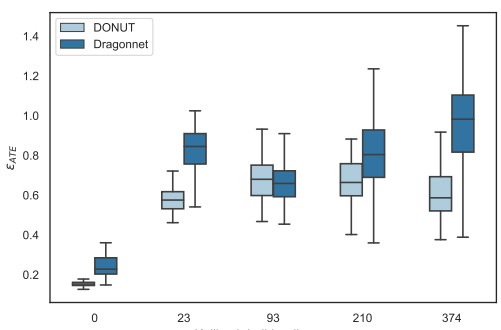

Figure 1: Comparison of DONUT and Dragonnet in presence of selection bias. Reported is the mean and standard deviation of $\epsilon_{\text{ATE}}$ on synthetic data as selection bias is varied (measured by the Kullback-Leibler divergence). For each level, both algorithms are run on 100 realizations of the datasets. Lower is better.

## 6 CONCLUSION

Understanding causal effects is crucial for reliable decision-making. In this paper, we present a regularization framework for estimating average causal effects. We formalize unconfoundedness as an orthogonality constraint that is used to learn outcomes that are orthogonal to the treatment assignment. We prove theoretical guarantees that our regularization framework yields an asymptotically normal estimator. Based on this, we develop DONUT, which leverages the predictive capabilities of neural networks to estimate average causal effects. Experiments on datasets show that, in most cases, DONUT outperforms the state-of-the-art substantially. This work provides an interesting avenue for future research on causal inference. We hypothesize that most existing models can be improved by

incorporating orthogonal regularization. We leave the derivation of a unifying theory for future work. A revised version of this work can be found in Hatt & Feuerriegel (2021).

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

## A    SUFFICIENCY OF REGULARIZING THE UNTREATED OUTCOME

We show that it is sufficient to regularize $Y(0)$, since orthogonality of $Y(0)$ implies orthogonality of $Y(1)$ to $T$ given $X$. Suppose $Y(0)$ satisfies the orthogonality constraint, i.e.,

$$\langle Y(0) - f(X,0), T - \pi(X) \rangle = \frac{1}{n} \sum_{i=1}^{n} (Y_i(0) - f(X_i, 0))(T_i - \pi(X_i)) = 0. \tag{14}$$

Then,

$$\langle Y(1) - f(X,1), T - \pi(X) \rangle = \frac{1}{n} \sum_{i=1}^{n} (Y_i(1) - f(X_i, 1))(T_i - \pi(X_i)) \tag{15}$$

$$= \frac{1}{n} \sum_{i=1}^{n} (Y_i(1) - \mathbb{E}[Y_i \mid X_i, T = 1])(T_i - \pi(X_i)) \tag{16}$$

$$= \frac{1}{n} \sum_{i=1}^{n} (Y_i(1) - \mathbb{E}[Y_i(1) \mid X_i])(T_i - \pi(X_i)) \tag{17}$$

$$= \frac{1}{n} \sum_{i=1}^{n} (Y_i + \psi(X_i)(1 - T_i) - \mathbb{E}[Y_i + \psi(X_i)(1 - T_i) \mid X_i])(T_i - \pi(X_i)) \tag{18}$$

$$= \frac{1}{n} \sum_{i=1}^{n} (Y_i - \psi(X_i) T_i - \mathbb{E}[Y_i - \psi(X_i) T_i \mid X_i])(T_i - \pi(X_i)) \tag{19}$$

$$= \frac{1}{n} \sum_{i=1}^{n} (Y_i(0) - \mathbb{E}[Y_i(0) \mid X_i])(T_i - \pi(X_i)) \tag{20}$$

$$= \frac{1}{n} \sum_{i=1}^{n} (Y_i(0) - \mathbb{E}[Y_i \mid X_i, T = 0])(T_i - \pi(X_i)) \tag{21}$$

$$= \frac{1}{n} \sum_{i=1}^{n} (Y_i(0) - f(X_i, 0))(T_i - \pi(X_i)) \tag{22}$$

$$= \langle Y(0) - f(X,0), T - \pi(X) \rangle = 0, \tag{23}$$

using that $Y(0) = Y - \psi(X) T$ and $Y(1) = Y + \psi(X)(1 - T)$ in (18) and (20). Hence, $\langle Y(1) - f(X,1), T - \pi(X) \rangle$ is zero and, therefore, $Y(1)$ is orthogonal to $T$ given $X$. Nevertheless, it is straightforward to adapt the orthogonal regularizer in (8) to accommodate both outcomes. Consider the following adapted orthogonal regularizer

$$\Omega_{OR}(f, \pi, \epsilon; \mathcal{D}) = \frac{1}{n} \sum_{i=1}^{n} (Y_i^*(0) - f^\epsilon(0, X_i))^2 + \frac{1}{n} \sum_{i=1}^{n} (Y_i^*(1) - f^\epsilon(1, X_i))^2, \tag{24}$$

with

$$Y_i^*(0) = Y_i - \psi^* T_i, \qquad Y_i^*(1) = T_i + \psi^* (1 - T_i), \quad \psi^* = \frac{1}{n} \sum_{i=1}^{n} f(X_i, 1) - f(X_i, 0) \tag{25}$$

and

$$f^\epsilon(X_i, 0) = f(X_i, 0) + \epsilon(T_i - \pi(X_i)), \tag{26}$$

$$f^\epsilon(X_i, 1) = f(X_i, 1) + \epsilon(T_i - \pi(x_i)), \tag{27}$$

where $\epsilon$ is a additional model parameters. Similar to the result in Section 3.2.2., this yields that $\langle Y^*(0) - f^\epsilon(X, 0), T - \pi(X) \rangle$ and $\langle Y^*(1) - f^\epsilon(X, 1), T - \pi(X) \rangle$ are zero, when the partial derivative of (24) w.r.t. $\epsilon$ is zero.

## B ASYMPTOTICALLY NORMAL ESTIMATORS

In this section, we give a brief discussion on asymptotically normal estimators. We refer to van der Vaart (1998) for an in-depth discussion. Suppose $\hat{\psi}$ is an estimator for the true parameter $\psi$. Then, $\hat{\psi}$ is an asymptotically normal estimator if it satisfies

$$\hat{\psi} - \psi = \frac{1}{n}\sum_{i=1}^{n}\phi(Z_i) + o_p(1/\sqrt{n}), \tag{28}$$

where $\phi(Z)$ is referred to as the influence function at $Z = (X, T, Y)$ (Kandasamy et al., 2015). The influence function has mean zero and finite variance (i.e., $\mathbb{E}[\phi(Z)] = 0$ and $\text{Var}(\phi(Z)) < \infty$). The existence and uniqueness of the influence function follows by the Riesz representation theorem. Asymptotic normality is a favorable property of an estimator, since, by the central limit theorem,

$$\sqrt{n}(\hat{\psi} - \psi) \xrightarrow{d} \mathcal{N}(0, \text{Var}(\phi(Z))). \tag{29}$$

Hence, an asymptotically normal estimator is asymptotically normal distributed and unbiased. Moreover, the variance shrinks in proportion to $1/\sqrt{n}$ as the sample size grows and is given by the variance of the influence function. As a consequence, the distribution of the estimator $\hat{\psi}$ converges weakly to a dirac delta function centered around the true $\psi$.

## C PROOF OF THEOREM 1

In observational studies, the covariates $X$ are often high-dimensional and limited knowledge about the nuisance functions (i.e., outcome and propensity score functions $f$ and $\pi$) is available. In such a case, it is reasonable to use flexible, data-adaptive methods for estimating the nuisance functions, e.g., neural networks. However, the complexity of these methods makes asymptotic analysis difficult, because the estimators used to construct $\hat{\eta} = (\hat{f}, \hat{\pi})$ are not described by a single finite-dimensional parameter. Nevertheless, under some conditions, we can learn about the asymptotics of $\hat{\psi}$ using tools from empirical process theory.

First, we need to introduce some notation. Throughout the proof, we will use $\mathbb{P}\{f(Z)\} = \int f(z)d\mathbb{P}$ to denote expectations of $f(Z)$ for a random variable $Z$ (treating the function $f$ as fixed). Hence, $\mathbb{P}\{\hat{f}(Z)\}$ is random if $\hat{f}$ is random (e.g., estimated from the sample). In contrast, $\mathbb{E}[\hat{f}(Z)]$ is a fixed non-random quantity, which averages over randomness in both $Z$ and $\hat{f}$ and thus will not equal $\mathbb{P}\{\hat{f}(Z)\}$ except when $\hat{f} = f$ is fixed and non-random. Moreover, we let $\mathbb{P}_n = \frac{1}{n}\sum_{i=1}^{n}\delta_{Z_i}$ denote the empirical measure such that sample averages can be written as $\frac{1}{n}\sum_{i=1}^{n}f(Z_i) = \int f(z)d\mathbb{P}_n = \widehat{\mathbb{P}}_n(f(Z))$. For clarity, we will denote the true nuisance functions and average treatment effect as $\eta_0 = (f_0, \pi_0)$ and $\psi_0$, respectively. We further denote the triplet $(Y, T, X)$ by $Z$.

For $\hat{f} = f^\epsilon$, $\hat{\pi}$, and $\hat{\psi} = \frac{1}{n}\sum_{i=1}^{n}f^\epsilon(X_i, 1) - f^\epsilon(X_i, 0)$ from (8), the regularization framework yields

$$\langle Y^*(0) - \hat{f}(X, 0), T - \hat{\pi}(X)\rangle = \frac{1}{n}\sum_{i=1}^{n}(Y_i - \hat{\psi}\, T_i - \hat{f}(X_i, 0))(T_i - \hat{\pi}(X_i)) = 0, \tag{30}$$

by construction. Hence, we receive an analytical expression of the estimator of the average treatment effect by solving the above for $\hat{\psi}$, i.e.,

$$\hat{\psi} = \frac{\frac{1}{n}\sum_{i=1}^{n}(Y_i - \hat{f}(X_i, 0))(T_i - \hat{\pi}(X_i))}{\frac{1}{n}\sum_{i=1}^{n}T_i(T_i - \hat{\pi}(X_i))} \tag{31}$$

Therefore, the estimator is given by $\hat{\psi} = \widehat{\mathbb{P}}_n(m(Z; \hat{\eta}))$, where

$$m(Z; \eta) = \frac{(Y - f(X, 0))(T - \pi(X))}{\mathbb{P}\{T(T - \pi(X))\}}, \tag{32}$$

and $\eta = (\pi, f)$ denotes the nuisance functions.

We recall the conditions in Theorem 1. Suppose the estimator $\hat{\eta} = (\hat{f}, \hat{\pi})$ converges to some $\bar{\eta} = (\bar{f}, \bar{\pi})$ in the sense that $\|\hat{\eta} - \bar{\eta}\| = o_p(1)$[8], where either $\bar{f} = f_0$ or $\bar{\pi} = \pi_0$ (or both) corresponds to the

---

[8] $o_p(1/r_n)$ employs the usual stochastic order notation so that $X_n = o_p(1/r_n)$ means that $r_n X_n \to 0$ in probability.

true nuisance function. Thus at least one nuisance estimator needs to converge to the correct function, but one can be misspecified. Then, $\mathbb{P}\{m(Z;\bar\eta)\} = \mathbb{P}\{m(Z;\eta_0)\} = \psi_0$, from the straightforward to check fact that $\mathbb{P}\{(m(Z;\bar\pi,f_0)\} = \mathbb{P}\{m(Z;\pi_0,\bar f)\}$ for any $\bar\pi$ and $\bar f$. Consider the decomposition

$$
\begin{aligned}
\hat\psi - \psi_0 &= \widehat{\mathbb{P}}_n\left(m(Z;\hat\eta)\right) - \mathbb{P}\{m(Z;\bar\eta)\} \\
&= (\mathbb{P}_n - \mathbb{P})\{m(Z;\hat\eta)\} + \mathbb{P}\{m(Z;\hat\eta) - m(Z;\bar\eta)\}.
\end{aligned}
$$

If the estimators for the nuisance functions $\hat\eta$ take values in $\mathbb{P}$-Donsker classes, then $m(Z;\hat\eta)$ also belongs to a $\mathbb{P}$-Donsker class, since Lipschitz transformations of Donsker functions are again Donsker functions. The Donsker property together with the continuous mapping theorem yields that $(\mathbb{P}_n - \mathbb{P})\{m(Z;\hat\eta)\}$ is asymptotically equivalent to $(\mathbb{P}_n - \mathbb{P})\{m(Z;\bar\eta)\}$ up to $o_p(1/\sqrt{n})$ error (see Mikosch et al. (1997) for more details on $\mathbb{P}$-Donsker classes). Therefore,

$$
\hat\psi - \psi_0 = (\mathbb{P}_n - \mathbb{P})\{m(Z;\bar\eta)\} + \mathbb{P}\{m(Z;\hat\eta) - m(Z;\bar\eta)\} + o_p(1/\sqrt{n}). \tag{33}
$$

It is left to show that $\mathbb{P}\{m(Z;\hat\eta) - m(Z;\bar\eta)\}$ is asymptotically negligible. This term equals

$$
\mathbb{P}\left\{ \frac{(T - \hat\pi(X))(Y - \hat f(X,0))}{\mathbb{P}\{T(T - \hat\pi(X))\}} - \frac{(T - \pi_0(X))(Y - f_0(X,0))}{\mathbb{P}\{T(T - \pi_0(X))\}} \right\} \tag{34}
$$

$$
= \mathbb{P}\left\{ \pi_0(X)\left( \frac{(1 - \hat\pi(X))(f_0(X,1) - \hat f(X,0))}{\mathbb{P}\{T(T - \hat\pi(X))\}} - \frac{(1 - \pi_0(X))(f_0(X,1) - f_0(X,0))}{\mathbb{P}\{T(T - \pi_0(X))\}} \right) \right\} \tag{35}
$$

$$
- \mathbb{P}\left\{ \frac{(1 - \pi_0(X))\hat\pi(X)(f_0(X,0) - \hat f(X,0))}{\mathbb{P}\{T(T - \hat\pi(X))\}} \right\} \tag{36}
$$

$$
= \mathbb{P}\left\{ \pi_0(X)\left( \frac{(1 - \hat\pi(X))(f_0(X,1) - \hat f(X,0))}{\mathbb{P}\{\pi_0(X)(1 - \hat\pi(X))\}} - \frac{(1 - \pi_0(X))(f_0(X,1) - f_0(X,0))}{\mathbb{P}\{\pi_0(X)(1 - \pi_0(X))\}} \right) \right\} \tag{37}
$$

$$
- \mathbb{P}\left\{ \frac{(1 - \pi_0(X))\hat\pi(X)(f_0(X,0) - \hat f(X,0))}{\mathbb{P}\{\pi_0(X)(1 - \hat\pi(X))\}} \right\} \tag{38}
$$

$$
= \mathbb{P}\left\{ \frac{\pi_0(X)(1 - \hat\pi(X))(f_0(X,1) - \hat f(X,0))}{\mathbb{P}\{\pi_0(X)(1 - \hat\pi(X))\}} \right\} - \psi_0 \tag{39}
$$

$$
- \mathbb{P}\left\{ \frac{(1 - \pi_0(X))\hat\pi(X)(f_0(X,0) - \hat f(X,0))}{\mathbb{P}\{\pi_0(X)(1 - \hat\pi(X))\}} \right\} \tag{40}
$$

$$
= \mathbb{P}\left\{ \frac{\pi_0(X)(1 - \hat\pi(X))(f_0(X,1) - \hat f(X,0))}{\mathbb{P}\{\pi_0(X)(1 - \hat\pi(X))\}} - \frac{\pi_0(X)(1 - \hat\pi(X))(f_0(X,1) - f_0(X,0))}{\mathbb{P}\{\pi_0(X)(1 - \hat\pi(X))\}} \right\} \tag{41}
$$

$$
- \mathbb{P}\left\{ \frac{(1 - \pi_0(X))\hat\pi(X)(f_0(X,0) - \hat f(X,0))}{\mathbb{P}\{\pi_0(X)(1 - \hat\pi(X))\}} \right\}, \tag{42}
$$

where we use $\mathbb{P}\{m(Z;\bar\eta)\} = \mathbb{P}\{m(Z;\eta_0)\}$ in (34), iterated expectation in (35), and $\mathbb{P}\{T(T - \hat\pi(X))\} = \mathbb{P}\{\pi_0(X)(1 - \hat\pi(X))\}$ (likewise for $\pi_0$ instead of $\hat\pi$) in (37). In (39), and similarly in (341), we use that $\psi_0(X) = \psi_0$ and, therefore, $f_0(X,1) - f_0(X,0) = \psi_0$. As a result, by simplifying, the above equals

$$
\frac{\pi_0(X) - \hat\pi(X)}{\mathbb{P}\{\pi_0(X)(1 - \hat\pi(X))\}}(f_0(X,0) - \hat f(X,0)). \tag{43}
$$

Therefore, by the fact that $\pi_0$ and $\hat\pi$ are bounded away from zero and one, along with the Cauchy-Schwarz inequality (i. e., $\mathbb{P}\{fg\} \le \|f\|\|g\|$), we have that (up to a multiplicative constant) $|\mathbb{P}\{m(Z;\hat\eta) - m(Z;\bar\eta)\}\}|$ is bounded above by

$$
\|\pi_0(X) - \hat\pi(X)\|\|f_0(X,0) - \hat f(X,0)\|. \tag{44}
$$

Thus, for example if $\hat\pi$ is based on a correctly specified parametric model (e. g., logistic regression), so that $\|\hat\pi - \pi_0\| = o_p(1/\sqrt{n})$, then we only need $\hat f$ to be consistent, i. e., $\|\hat f - f_0\| = o_p(1)$, to make the product term $\mathbb{P}\{m(Z;\hat\eta) - m(Z;\bar\eta)\} = o_p(1/\sqrt{n})$ asymptotically negligible. Then the doubly robust estimator satisfies $\hat\psi - \psi_0 = (\mathbb{P}_n - \mathbb{P})\{m(Z;\eta_0)\} + o_p(1/\sqrt{n})$ and it is efficient with

influence function $\phi(Z; \psi, \eta) = m(Z; \eta) - \psi$. Thus, this proves the asymptotic normality of $\hat{\psi}$. It remains to show that the asymptotic variance of $\hat{\psi}$ equals

$$\frac{\sigma^2}{\mathbb{P}\{\text{Var}(T \mid X)\}}. \tag{45}$$

Note that in the paper, $\mathbb{E}[f(Z)]$ is used similarly to $\mathbb{P}\{f(Z)\}$. We introduced the notion $\mathbb{P}\{f(Z)\}$ here to make clear that we only integrate over randomness in $Z$ and not the function estimates $\hat{f}$. As seen in Appendix B, the asymptotic variance of an estimator is the variance of its influence functions. From above, we know that the efficient influence function of $\hat{\psi}$ is $\phi(Z; \psi, \eta) = m(Z; \eta) - \psi$. Thus, using that $\mathbb{P}\{\phi(Z; \psi, \eta)\} = 0$ (by definition),

$$\text{Var}(\phi(Z; \psi, \eta)) = \mathbb{P}\{\phi(Z; \psi, \eta)^2\} \tag{46}$$

$$= \mathbb{P}\{(m(Z; \eta) - \psi)^2\} \tag{47}$$

$$= \mathbb{P}\{(m(Z; \eta)^2\} - \psi^2. \tag{48}$$

This, together with

$$\mathbb{P}\{m(Z; \eta)^2\} = \frac{\mathbb{P}\{U^2\}}{\mathbb{P}\{\text{Var}(T \mid X)\}} + \psi^2, \tag{49}$$

where $U = Y - f(X, T) \sim \mathcal{N}(0, \sigma^2)$ and $\mathbb{P}\{T(T - \pi(X))\} = \mathbb{P}\{\text{Var}(T \mid X)\}$ proves the statement. $\qquad\square$

## D   ORTHOGONAL REGULARIZATION UNDER LINEAR SPECIFICATION OF THE TREATMENT EFFECT

In Theorem 1, condition 2 ensures the homogeneity of the treatment effect, i. e., $\psi(X) = \psi$. This can be easily relaxed to any specification of $\psi$ as long as it has finitely many parameters and given that the appropriate identification criteria hold (for linear specification, this is the non-singularity of the design matrix). We explain in this section how the orthogonal regularization in (8) changes when linear specification of the treatment effect is considered, i. e., $\psi(x) = \theta^\top x$, where $\theta \in \mathbb{R}^d$. Recall the orthogonal regularizer from (8)

$$\Omega_{OR}(f, \pi, \epsilon; \mathcal{D}) = \frac{1}{n} \sum_{i=1}^n (Y_i^*(0) - f^\epsilon(X_i, 0))^2, \tag{50}$$

where the perturbation function $f^\epsilon$ remains unchanged. The pseudo outcome $Y_i^*(0)$ is now given as

$$Y_i^*(0) = Y_i - (\theta^\top X_i) T_i, \tag{51}$$

where $\theta$ is given by the system of linear equations,

$$\theta^\top X_i = \frac{1}{n} \sum_{i=1}^n v(T_i, X_i)(f(X_i, 1) - f(X_i, 0)), \tag{52}$$

where $v(t, x)$ is an arbitrary function of the dimension of $\theta$, which ensures that the system of linear equations possesses a unique solution. For instance, in the case of $\psi(x) = \theta^\top x$, the choice $v(t_i, x_i) = x_i t_i$ could be made. The procedure goes as follows. First, the above system of linear equations is solved for $\theta$. Second, the pseudo outcome $Y_i^*(0) = Y_i - (\theta^\top X_i) T_i$ can be computed. Finally, this is plugged into the regularizer. The proof for asymptotic normality goes similar to the one under homogeneous treatment effect.

## E   FEEDFORWARD NEURAL NETWORKS ARE IN $\mathbb{P}$-DONSKER CLASS

We show that feedforward neural networks are in a $\mathbb{P}$-Donsker class. A feedforward neural network $f_d$ is defined by the recursion

$$f_1(x) = \mathbf{W_1}^\top x, \qquad f_i(x) = \mathbf{W_i}^\top \sigma(f_{i-1}(x)), \quad i = 2, \dots, d, \tag{53}$$

for $d \in \mathbb{N}_{\geq 1}$, matrices $\{\mathbf{W_i}\}_{i=1}^{d}$ of appropriate dimensions, activation function $\sigma$, which applies element-wise, and $d$ is the depth of the neural network.

$f_1$ is clearly in a $\mathbb{P}$-Donsker class as it is Lipschitz parametric. The Donsker property is preserved under Lipschitz transformations. Hence, $\hat{f}_i(x)$, for $i = 2, \ldots, d$, is in a $\mathbb{P}$-Donsker class, if the activation function is a Lipschitz transformation. This is the case for most common activation functions. We give two examples in the following.

The ReLU activation function, $\sigma_{\text{ReLU}}(x) = \max(0, x)$, preserves the Donsker property, since a constant function is clearly in a $\mathbb{P}$-Donsker class and taking the maximum of two $\mathbb{P}$-Donsker functions is again a $\mathbb{P}$-Donsker function due to the preservation under Lipschitz transformations. The same follows for the ELU activation function,

$$\sigma_{\text{ELU}}(x) = \left\{ \begin{array}{ll} x, & \text{if } x > 0 \\ \alpha(\mathrm{e}^x - 1), & \text{if } x \leq 0 \end{array} \right.,$$

where $\alpha \geq 0$, since it is a Lipschitz transformation.

## F  ASYMPTOTICS OF $\hat{\psi}$ UNDER HETEROGENEOUS TREATMENT EFFECTS

Our estimator retains a useful interpretation under heterogeneous treatment effects. In this case, $\hat{\psi}$ converges to

$$\frac{\mathbb{P}\{\text{Var}(T \mid X)\psi(X)\}}{\mathbb{P}\{\text{Var}(T \mid X)\}}, \tag{54}$$

where $\psi(X) = f(X, 1) - f(X, 0)$ is the true treatment effect at $X$. This can be interpreted as a weighted average of treatment effects $\psi(X)$. As a consequence, most weight is given to subpopulations with large $\text{Var}(T \mid X)$, i. e., which are most informative about the treatment effect.

We consider the same notation as in the proof in Appendix C and resume it. From the proof in Appendix C, we know that

$$\hat{\psi} - \psi = (\mathbb{P}_n - \mathbb{P})\{m(Z; \bar{\eta})\} + \mathbb{P}\{m(Z; \hat{\eta}) - m(Z; \bar{\eta})\} + o_p(1/\sqrt{n}), \tag{55}$$

since the estimator of the nuisance functions $\hat{\eta} = (\hat{f}, \hat{\pi})$ belong to a $\mathbb{P}$-Donsker class, and therefore $m(Z; \hat{\eta})$ belongs to a $\mathbb{P}$-Donsker class. Similar to the proof in Appendix C, it is left to investigate the term $\mathbb{P}\{m(Z; \hat{\eta}) - m(Z; \bar{\eta})\}$. We show that this term is asymptotically not negligible when the treatment effect is heterogeneous. Again, by iterated expectations, this term equals

$$\mathbb{P}\left\{ \frac{(\pi_0(X) - \hat{\pi}(X))(f_0(X, 1) - \hat{f}(X, 0))}{\mathbb{P}\{\pi_0(X)(1 - \hat{\pi}(X))\}} \right\} - \mathbb{P}\left\{ \frac{\pi_0(X)(1 - \pi_0(X))(f_0(X, 1) - f_0(X, 0))}{\mathbb{P}\{\pi_0(X)(1 - \pi_0(X))\}} \right\}. \tag{56}$$

The first term can be bounded from above (up to a multiplicative constant) by the Cauchy-Schwarz inequality and, therefore, is asymptotically negligible if one of the nuisance functions is correctly specified. Thus, only the second term remains. Using $\psi(X) = f_0(X, 1) - f_0(X, 0)$, the second term can be written as

$$\frac{\mathbb{P}\{\pi_0(X)(1 - \pi_0(X))\psi(X)\}}{\mathbb{P}\{\pi_0(X)(1 - \pi_0(X))\}}. \tag{57}$$

Since $\pi_0(X)(1 - \pi_0(X)) = \text{Var}(T \mid X)$, this concludes the proof.  □

## G  COMPARISON TO INVERSE PROBABILITY WEIGHTED ESTIMATOR

We compare our estimator $\hat{\psi}$ to the inverse probability weighted (IPW) estimator (e. g., Funk et al., 2011). The IPW estimator is a popular estimator for treatment effects due to its improvement in efficiency and reduction in bias compared to unweighted estimators.

Under identical conditions as in Theorem 1, the IPW estimator, denoted as $\hat{\psi}_{\text{IPW}}$, is asymptotically normal with asymptotic variance $\sigma^2 \mathbb{E}[1/\text{Var}(T \mid X)]$ (e. g., Kennedy, 2016). We can compare the asymptotic behavior of our estimator $\hat{\psi}$ and the IPW estimator $\hat{\psi}_{\text{IPW}}$. Both estimators are asymptotically unbiased, but with different asymptotic variance as the following result shows.

**Corollary 1.** *Under the conditions in Theorem 1, the asymptotic variance of $\hat{\psi}$ is strictly smaller than the asymptotic variance of $\hat{\psi}_{IPW}$.*

*Proof.* The statement follows by Jensen's inequality. □

The difference between the asymptotic variances becomes particularly pronounced in presence of selection bias, i.e., when the treatment group is systematically different from the control group.

**Corollary 2.** *If the propensity score $\pi(x)$ is close to 0 or 1 for some $x \in \mathcal{R}^d$, then the asymptotic variance of $\hat{\psi}_{IPW}$ can take arbitrary large values.*

*Proof.* For some small $\epsilon > 0$, let $\pi(X) \in [0,1]\backslash(\epsilon, 1-\epsilon)$. Then, by Bhatia-Davis inequality and since $T$ is binary, $\mathrm{Var}(T \mid X) = \pi(X)(1 - \pi(X))$. Hence, $\mathrm{Var}(T \mid X) \leq \epsilon$, and, therefore, $\mathbb{E}[1/\mathrm{Var}(T \mid X)] \geq 1/\epsilon$, which proves the statement. □

## H    DETAILED DESCRIPTION OF THE DATASETS

### H.1    IHDP

Hill (Hill, 2011) introduced a semi-synthetic dataset created from the Infant Healthand Development Program (IHDP). This dataset is based on a randomized experiment that examines the effect of home visits by specialists on future cognitive scores. The dataset consists of 747 children ($t = 1$: 139, $t = 0$: 608) with 25 covariates. Similar to Shalit et al. (2017), we use 1,000 realizations from setting A in the NPCI package (Dorie, 2016).

### H.2    TWINS

This dataset is made up of all births in the USA between 1989 and 1991 (Almond et al., 2005). Only twins are considered among these births. Treatment (i.e., $T = 1$) is defined as being the heavier twin (and $T = 0$ as being the lighter twin). The outcome is defined as 1-year mortality. There are 30 covariates available for each pair of twins that relate to the parents, pregnancy and birth: marital status; race; residence; number of previous births; pregnancy risk factors; quality of care during pregnancy; and number of gestation weeks prior to birth. Only twins that weigh less than 2 kg and have no missing covariates (list-wise deletion) are taken into account. This creates a complete dataset (without missing data).[9] The final cohort consists of 11,400 twin pairs, whose mortality rate is 17.7 % for lighter twins and 16.1 % for heavier ones. In this setting, we observed both $T = 0$ (lighter twin) and $T = 1$ (heavier twin) for each pair of twins; Therefore, the true treatment effect in this dataset is known. In order to simulate an observational study, one of the twins is selectively observed based on information using the covariates (which leads to selection bias) as follows: $T \mid x \sim \mathrm{Bern}(\mathrm{Sigmoid}(w^\top x + n))$ where $w^\top \sim \mathcal{U}((-0.1, 0.1)^{30 \times 1})$ and $n \sim \mathcal{N}(0, 0.1)$.

### H.3    JOBS

Jobs studied in LaLonde (1986) consists of randomized data based on the National Supported Work program and non-randomized observational study data. A (random) subset of randomized data is used to evaluate the algorithms. The dataset consists of 722 randomized samples ($T = 1$: 297, $T = 0$: 425) and 2,490 non-randomized samples ($T = 1$: 0, $T = 0$: 2,490), all with 7 covariates.

### H.4    ACIC 2018

ACIC 2018 is a collection of semi-synthetic datasets derived from the linked birth and infant death data (LBIDD) (MacDorman & Atkinso, 1998), and was developed for the 22018 Atlantic Causal Inference Conference competition (ACIC) (Shimoni et al., 2018). The simulation includes 63 different data generation processes with a sample size from 1,000 to 50,000. Each dataset is a realization from a separate distribution, which itself is randomly drawn in accordance with the settings of the data generation process. Similar to Shi et al. (2019), we randomly pick 3 datasets[10] of size either 5k or

---

[9]We provide the complete dataset at `github.com/anonymous/donut` (anonymized for peer-review).
[10]We provide the list of unique dataset identification numbers at `github.com/anonymous/donut` (anonymized for peer-review) for reproducibility.

10k for each of the 63 data generating process settings and exclude all datasets with indication of strong selection bias. This yields a total number of 97 datasets.

## I    DATA GENERATING PROCESS FOR SIMULATION STUDY

For the simulation study, we follow a similar protocol as in Yao et al. (2018); Yoon et al. (2018). We generate 2,500 untreated samples from $\mathcal{N}(\mathbf{0}^{10 \times 1}, 0.5 \times \Sigma\Sigma^\top)$ and 5,000 treated samples from $\mathcal{N}(\mu_1, 0.5 \times \Sigma\Sigma^\top)$, where $\Sigma \sim \mathcal{U}((0,1)^{10 \times 10})$. Varying $\mu_1$ yields different levels of selection bias, which is measured by the Kullback-Leibler divergence. The larger the Kullback-Leibler divergence, the greater the distributional distance between treatment and control group, and, thus, the larger the selection bias. The outcome is generated as $Y \mid X = \mathbf{x}, T = t \sim (\mathbf{w}^\top \mathbf{x} + t + n)$, where $\mathbf{w} \sim \mathcal{U}((-1,1)^{10 \times 1})$, and $n \sim \mathcal{N}(0, 0.1)$.

