# OpenReview forum: "Estimating Treatment Effects via Orthogonal Regularization"
_ICLR.cc/2021/Conference — Reject_

### Official Review · AnonReviewer4 · 2020-10-28
**Interesting novel approach to combine deep learning and causal inference by leveraging implications from the classical unconfoundedness assumption.**

**Rating:** 7
**Confidence:** 4

**Review:**

Summary:
The present paper introduces a new approach, deep orthogonal networks for unconfounded treatments (DONUT), that allows to estimate (average) treatment effects exploiting an orthogonality property implied by the classical unconfoundedness assumption. The authors propose a regularization framework based on the orthogonality constraint and prove that a resulting estimator is doubly robust, asymptotically normal and with efficient variance. They supply multiple simulations to demonstrate their theoretical claims and to show state-of-the-art performance of their estimator.

Recommendation:
Clear accept. In summary, I am convinced that the this paper would be a valuable addition to this year's conference. It considers a novel approach to improve average treatment effect estimation on observational data using combining classical causal inference assumptions and predictive power of deep learning.

Strong points:
 - The authors propose a new methodology that seems theoretically solid and that has an implementable estimator for ATE estimation, exploiting a necessary condition implied by a standard causal inference assumption.
 - The article is well written and easy to read.

Weak points:
 - The code for their simulations is not accessible (broken/incorrect url?).
 - A discussion about the impact of the hyperparameters, especially the orthogonality regularization parameter $\lambda$, would give more insight into the importance of the contribution of the regularization term.

Questions/Issues:
 - The provided url to access the anonymized code did not work (at least for me), would it be possible to fix this or to provide the complete code as supplementary zip file?
 - How sensitive are the results to the hyperparameter choices, especially the $\lambda$ parameter?
 - The orthogonality constraint is implied by the unconfoundedness assumption, but it is not a sufficient condition for unconfoundedness. Have the authors studied the behaviour of their method and its performance in (simulated) cases where unconfoundedness (4) does not hold but the orthogonality constraint (5) holds?
 - The authors mention the R-learner (Nie \& Wager, 2017) which uses the notion of Neyman orthogonality and use the R decomposition to propose an estimator of treatment effects. Would it be possible to add this method to the list of compared methods?
 - Since the authors theoretically compare their estimator to the IPW estimator, it would be interesting to add this to the experiments to confirm the theoretical results.
 - For the ACIC datasets, how does DONUT compare to BART, which is known to perform well on these data (Dorie et al., 2018)?

Minor comments (that did not impact the score):
 - p. 2/3: equations (2) and (4) are the same.
 - p. 5: yield by our $>>$ yielded by our

References:
 - Dorie V, Hill J, Shalit U, Scott M, Cervone D. Automated versus do-it-yourself methods for causal inference: Lessons learned from a data analysis competition. Statistical Science. 2019;34(1):43--68.
 - Nie X, Wager S. Quasi-oracle estimation of heterogeneous treatment effects. arXiv preprint arXiv:1712.04912, 2017.

===================

Post Rebuttal Update:

After the discussions and reading the other reviews, I lower my score by one point. I could not find the changes to the manuscript announced by the authors during the discussion, especially the additional intermediate results necessary for the theorem's proof pointed out by reviewer 1.

---

> ### Author Response · Authors · 2020-11-13
> **RE: Interesting novel approach to combine deep learning and causal inference by leveraging implications from the classical unconfoundedness assumption.**
>
> We highly appreciate your feedback and comments. The following points clarify your questions.
> 1) The link to the github profile is anonymized to allow for a blind review. It will be accessible after the paper has been published. In the meantime, the code for DONUT can be found in the supplements.
> 2) $\lambda$ in the orthogonal regularizer (8) is determined by hyperparameter optimization over $(10^k)_{k=-2}^2$. For very small value of $\lambda$, the results become similar to those of TARNet. This is reasonable, since TARNet is DONUT without the regularizer. As such, DONUT "approaches" TARNet for small $\lambda$, since the influence of the orthogonal regularizer becomes small. Hence, $\lambda$ has to be chose appropriately, but once $\lambda$ passes a certain threshold, we observed, for our datasets, that the results are stable for different $\lambda$. We will include this in a revised version of the manuscript.
> 3) We did not run a simulation study that studies the behaviour of DONUT in the case that unconfoundedness is violated, but the orthogonality constraint holds. The reasons for this is that DONUT already performs superior across the benchmarks datasets.
> 4) We have not compared to the R-learner in our experiments. However, we compared to partial linear regression (PLR) and Dragonnet, which are both based on Neyman orthogonality.
> 5) For preciously this reasons, we compared DONUT to Dragonnet in the simulations study on selection bias (Section 5.3). Dragonnet is based on the augmented inverse probability weighted estimator (AIPW), which is an augmented version of the IPW estimator. We observe what is expected from our theoretical results: for larger selection biases, the variance of DONUT is smaller than the variance of Dragonnet. See Section 5.3 and Figure 1 in the paper for more details.
> 6) We did not run BART on the ACIC 2018 dataset collection. We only compared DONUT to Dragonnet and TARNet. The reason for this is twofold: (i) We wanted to compare our method to the current state-of-the-art, Dragonnet, and show that DONUT is superior not only on IHDP, TWINS, and Jobs, but on a wide range of different datasets. (ii) We wanted to compare DONUT to TARNet as an ablation analysis on the orthogonal regularizer, since TARNet is DONUT, but without the orthogonal regularizer $\Omega_{OR}$. Hence, this shows the substantial contribution of the orthogonal regularizer to the performance. We will clarify this in a revised version of the manuscript.
>
> Response to "Minor comments":
> 7) We repeated the equation for ease of readability.
> 8) We will correct this. Many thanks.

---

### Official Review · AnonReviewer2 · 2020-10-29
**The motivation and some parts about the method are not clear**

**Rating:** 5
**Confidence:** 4

**Review:**

The authors propose a regularized framework for estimating the average treatment effect. They assume unconfoudedness and show that it implies a specific orthogonality constraint. The main idea is to use this orthogonality constraint during estimation of the model parameters as a regularizer. On the theoretical side, the authors provide sufficient conditions under which the regularization yields an asymptotically normal estimator for the average causal effect. Based on the regularization framework, an estimator for average causal effect via feedforward neural nets is developed.

- The motivation for the regularization is not clear. It is not clarified in the paper that why the proposed regularization should in fact improve the estimation bias and variance. It is only shown that the resulting estimator is asymptotically normal (the authors also show efficiency but it is under the strong assumption 2 in the theorem), but it is not clear that why this regularization would be particularly helpful.

- Page 4, the authors say "if we had access to the treatment effect \psi(X)=f(X,1)-f(X,0), we would also have access to the untreated outcome Y(0) even if we did not observe it. This does not seem to be true. (same for the footnote) f(X,0) is a function of X and if we take its expected value, under positivity, consistency and ignorability, it gives us the expected value of Y(0) not the random variable itself.

- Then \psi(X) is replaced by its empirical mean and the authors say "Under sufficient conditions, this converges to the true outcome (see Section 4)". This does not seem to be what is proven in Theorem 1.

- How can we show that equation 11 implies equation 5, which was the original goal?

- Assumption 2 in Theorem 1 is very strong and I suggest removing it from the manuscript.

---

> ### Author Response · Authors · 2020-11-13
> **RE: The motivation and some parts about the method are not clear**
>
> Thank you for your valuable feedback. In the following, we provide clarifying answers to your questions.
> 1) The motivation of the regularization is as follows. In existing work, unconfoundedness (Eq. (2)) is only used to identify the causal effect. However, unconfoundedness provides us with more information about the potential outcomes than that. In particular, we know from unconfoundedness that when we estimate the potential outcomes, the resulting estimates have to be conditionally uncorrelated (i.e., orthogonal with respect to the inner product in (5)). We encode this additional information in the orthogonality constraint, which is then used in the estimation procedure of the model parameters.
> 2) In the description of the pseudo outcome on page 4, the identities are meant "in expectation". We will clarify this in the revised version of the manuscript.
> 3) As above, the identities are meant "in expectation". The convergence of the pseudo outcome can be seen as follows. Under the conditions of Theorem 1, $\psi^\ast$ converges to the true ATE $\psi$. Therefore, under the conditions of Theorem 1, the expectation of the pseudo outcome $Y_i^\ast(0) = Y_i - \psi^\ast T_i$ converges to the true outcome $Y_i(0)$.
> 4) This can be seen directly when the pseudo outcome is replaced by the true outcome (which is justified by the convergence of the pseudo outcome to the true outcome). Hence, equation 5 follows.

---

### Official Review · AnonReviewer1 · 2020-10-29
**Very interesting idea, but some technical issues**

**Rating:** 3
**Confidence:** 4

**Review:**

Overall, I found the main idea of this paper very interesting and the experimental results promising; however, there were several major and minor technical issues with the work that need to be resolved.

--- Major comments ---

1. There appear to be at least two major technical issues:

    a. A substantial portion of the work is based on the author's assertion that $Y_i(0) = Y_i - \psi(X_i)T_i$ which is not, in general, true. We can see this with a simple counter-example. Let $Y(0)$ be a binary variable. Then $\psi(X) = E[Y(1)|X] - E[Y(0)|X]$ will be some value in $[-1,1]$, but if $\psi$ is any value other than $1$, $0$, or $-1$, then $Y_i - \psi(X_i)T_i$ will be non-binary and thus not equal to $Y_i(0)$. This assertion only holds if $X$ and $T$ uniquely determine $Y$, which is not generally the case. Thus, it remains to be shown that enforcing equation (8) is equivalent to enforcing equation (5).

    b. In the first condition of Theorem 1, the authors assume that either $\hat{f}$ or $\hat{\pi}$ is consistent. If $\hat{f}$ and $\hat{\pi}$ were estimated separately and used as plug-in estimators, as in Chernozhukov et al. (2018), this would be a reasonable assumption, as it would be up to the user to show that their nuisance parameter estimates are consistent. However, $\hat{f}$ and $\hat{\pi}$ are estimated using Equation (6) and thus it is up to the authors to show that solving (6) gives consistent estimates of $\hat{f}$ or $\hat{\pi}$ under correct model specification. This very well may be the case, but it cannot simply be assumed.

2. I found the introduction very hard to follow. In particular, it is never really made clear what the motivation for the work is. Extrapolating from the experiments, it appears that the goal is to derive an estimator with lower variance than existing estimators, but that is not stated in the intro. I would recommend restructuring to something like: CI from observational data is important because XYZ. It is desirable that estimators for the ATE have the lowest possible variance. We propose a new estimator that, empirically, has lower variance than a variety of state-of-the-art causal estimation methods. We do this by translating the exchangeability assumption into an explicit constraint on the estimator objective. This constraint reduces variance because ABC. Additionally, lit review portion of the intro reads as a random list of methods with no clear connection between them or to the proposed method. Some specific issues are:

    a. It is incorrect that "the outcome of an alternative treatment has to be estimated". In particular, IPW methods do not do this. In fact, methods based on estimating the conditional expected outcome do not do this either since, as stated above, the expected outcome is not equal to the outcome.

    b. "to find similar subjects" --> "to find similar subjects who received different treatments"

    c. "seek weights such that the treatment assignment is unassociated with the covariates" --> "reweights the data so that treatment assignment is unassociated with the covariates in the reweighted distribution"

    e. "However, they do not require the...": None of the methods described here require this, but they do require that they be conditionally independent. Also, potential outcomes have not yet been defined.

3. Page 2, Contribution section, "Compared to other estimators, its asymptotic variance is strictly smaller.": It is my understanding that TMLE is also semi-parametrically efficient, so I believe this statement is incorrect as are similar statements in the "Comparison to other estimators" section. Further, both Chernozhukov et al. (2018) and the work on CV-TMLE show $\sqrt{N}$-consistency without relying the Donsker class assumptions.

--- Minor comments ---

1. In equations (4) and (5), the authors jump from a constraint on the true distribution to a constraint on the empirical distribution. I recommend adding a statement indicating this.

2. What score was used to select the best $\lambda$ on the validation set?

3. RHS of Equation (11): $\epsilon$ --> $\hat{\epsilon}$ or drop the "evaluated at" bar.

---

> ### Author Response · Authors · 2020-11-13
> **RE: Very interesting idea, but some technical issues**
>
> Many thanks for your valuable comments. We provide answers to your questions below.
> 1a.)
> We would kindly clarify that the identities on page 4 are meant "in expectation". This might have been confusing and we will clarify this in the revised version of the manuscript. If the identities are understood in expectation, then the reviewer’s argument works, since $\mathbb{E}[Y(0)]$ does not need to be binary.
> 1b.) We provide a general, flexible framework for orthogonalization. The desired model specification is up to the user, whereby the user can choose the model specification of the $\hat{f}$ and $\hat{\pi}$. For instance, the user can choose whether to use neural networks for $\hat{f}$ and a logistic regression for $\hat{\pi}$ as we did for DONUT. Whether $\hat{f}$ and $\hat{\pi}$ converge in the sense that $\lVert \hat{\eta} - \bar{\eta} \rVert = o_p(1)$ under correct model specification has to be considered as well. In Theorem 1, we show which conditions are necessary such that the estimator is asymptotically normal. The components of the factual loss in (6) can be exchanged by the user if desired. However, we argue that all components in the loss function are either mean squared errors or cross entropies, which are standard. The components in the regularizer include a mean squared error of the pseudo outcome, which under sufficient conditions converges to the true potential outcome. Overall, this gives a powerful framework based on which a large class of models can be adapted to orthogonal regularization.
>
> 2.) The objective of the paper is to estimate the average treatment effect (ATE). We show that our estimator has lowest asymptotic variance. This, however, is not the main objective of the paper, but provides theoretical foundations of our estimator.
>
> We will revise the following writing issues pointed out as follows:
> 2a.) "the expected outcome of an alternative treatment has to be estimated"
> 2b.) "to find similar subjects who received the opposite treatment"
> 2c.) "seek weights to reweight the outcomes such that the treatment assignment is unassociated with the covariates in the reweighted distribution"
> 2e.) This comment only related to the work of Fong et al. (2018) and Yiu & Su (2018), not to the work described prior to Fong et al. (2018) and Yiu & Su (2018).
>
> 3.) We state in our paper that "under the conditions of Theorem 1, there does not exist a regular estimator that achieves strictly smaller asymptotic variance than our estimator". This is the case since our estimator achieves the lower efficiency bound as a result of its efficiency. As a consequence, other estimators (e.g., TMLE) might achieve the same asymptotic variance, but no other estimator achieves strictly smaller asymptotic variance than our estimator. In addition, it is not proven whether the standard TMLE is efficient under the same conditions as in Theorem 1.

---

> > ### Comment · AnonReviewer1 · 2020-11-22
> > **RE: Very interesting idea, but some technical issues**
> >
> > Thank you for the detailed response. I’ll elaborate on a few points:
> >
> > 1a) Clarifying this will help, though, rather than say “in expectation”, I would recommend simply writing the correct equality, that is: $E[Y(0)] = E[Y] - E[\psi(X)T]$. However, my primary concern is that the paper uses $Y(0) = Y - \psi(X)T$ as if it hold exactly. For example, in Equations 17-20. It is not clear to me that these equalities hold using only $E[Y(0)] = E[Y] - E[\psi(X)T]$ which holds only for the true distribution and not the empirical distribution.
> >
> > 1b) It seems I didn't communicate my concern clearly enough. I am going to try to be as clear as possible, and I apologize if some of this is obvious. Correct model specification is not the same as convergence in probability. As the authors pointed out in their response, if the model for $\hat{f}$ is correctly specified, $\hat{f}$ is Donsker, and we minimize squared error, then $\hat{f}$ should converge in probability to $f_0$, but that doesn’t necessarily hold for regularized squared error, which is what the authors are optimizing. Thus even if the model for $\hat{f}$ is correctly specified, the authors cannot simply assume convergence to $f_0$ (similarly for $\hat{\pi}$). I don’t necessarily think that this will be hard to prove (e.g. I think one can show that if the model for $\hat{f}$ is correctly specified, $\epsilon$ converges to zero and thus the regularizer converges to squared error and $\hat{f}$ converges to $f_0$), but I do believe it needs to be shown.
> >
> > 3) Two points: First, the statement included in the response is correct, but different from the statement I cited in my comment, which I think needs to be corrected. Second, I believe the conditions in this paper, specifically Conditions 1 and 3 of Theorem 1, are exactly the conditions under which TMLE is efficient (see, e.g., van der Laan and Starmans 2014).

---

> > > ### Author Response · Authors · 2020-11-23
> > > **RE: RE: Very interesting idea, but some technical issues**
> > >
> > > We thank you for the thoughtful comments and valuable feedback.
> > >
> > > 1a.) We apologize for the confusion and clarify this in the revised version of the manuscript. Indeed, it is correct that the derivation in Appendix A holds only true for the expected value (and, hence the conditional covariance) and not for the empirical mean (hence, the empirical conditional covariance). However, for $n \rightarrow\infty$ the expression in Appendix A converges to the expected value (by convergence of the mean to the true expected value). We will address this in the revised version of the manuscript.
> > >
> > > 1b.) Theorem 1 states the conditions under which our estimator for the ATE converges to the true ATE (at a fast rate). Condition 1 requires that either $\hat{f}$ or $\hat{\pi}$ converges to the true function. We agree that this needs to hold true within our optimization framework. We will state this in the paper and we will add a prove that $\hat{f}$ (or $\hat{\pi}$) converges in probability to the true $f_0$ (or $\pi_0$) under correct model specification in our optimization framework.
> > >
> > > 3.) Indeed, the TMLE is efficient under Conditions 1 and 3, but not if we add Condition 2. The TMLE uses the Augmented Inverse Probability Weighted (AIPW) estimator to estimate the ATE. This estimator is efficient under (what we state in our paper as) Conditions 1 and 3. However, looking at the asymptotic variance of the APIW estimator (p. 13 below Eq. 1 in van der Laan and Starmans 2014) reveals, that our estimator (under Condition 2) achieves smaller asymptotic variance. This is, however, at the price of stronger assumptions (Condition 2: homogeneous treatment effect).

---

### Official Review · AnonReviewer3 · 2020-11-04
**A novel method that performs admirably; some clarification needed in places**

**Rating:** 5
**Confidence:** 3

**Review:**

This paper proposes a novel regularization term for designing loss functions to estimate outcome and propensity score models, where the end goal is to estimate ATE.  The regularizer is derived from the assumption of conditional independence of potential outcomes and treatment given covariates (i.e. the no hidden confounding assumption).  The authors observe that this assumption implies that residuals of potential outcomes and treatments are orthogonal.  The authors derive a loss function which yields this orthogonality at the optimum.

The Theorem 1 shows asymptotic normality and a double robust property. The authors also perform extensive empirical comparisons to other causal inference methods on 4 standard benchmarks, as well as on the ACIC challenge. The method performs admirably: it is competitive on IHDP, and in a statistical tie with the best-performing methods on Twins and Jobs.

Unconfoundedness is an assumption that ensures that an estimator of the form of equation (3) is unbiased for a causal effect. Without the assumption, an estimator may ascribe a causal effect to the treatment that really comes from a common cause.  To use it to inform a penalty on the estimator is an unusual step, as it seems unrelated to the question of efficiency.  However, the results are compelling.

The first question that arises is: have the authors presented a clear ablation analysis, similar to the TARNet vs CFR analysis of Shalit et al.?  The authors state that they use the TARNet architectures for the outcome models.  In that case, can we interpret DONUT vs TARNet results in Table 1 as an ablation analysis on the $\Omega_{OR}$ regularizer? The numbers for TARNet in this submission are identical to those in Shalit et al., so perhaps not all experimental variables have been exactly matched such that we can consider it a true ablation analysis?

I have some technical questions that I believe the authors should address:
*    Moving from equation (31) to (32) seems to require omitting any randomness in the denominator.  Why is this justified?
*    Since $\hat f = f^{\epsilon}$ in equation 31, can we ever expect that $\bar f$ is equal to $f_0$?
*   What are the numbers in table 2?  The authors state they evaluate on 97 models from the ACIC challenge. Is table 2 presenting an average of ATE estimation errors?
*   In describing the Pseudo Outcomes, the authors state that $\psi(x)$ is the true treatment effect at X.  However, it is the expected treatment effect, and hence can’t be used to exactly impute an unobserved outcome, which may have some noise in general.  This also affects the argument in Appendix A (steps 18 and 20), where this imputation is performed.  Are the authors assuming throughout that the treatment effect is deterministic given $X$?  In Theorem 1, a constant treatment effect is assumed, but this seems to be an assumption to make the theory go, and the method is designed to handle heterogenous treatment effects.  If such an assumption is at play in the entire paper, it should be presented earlier.
*   Can the authors address how to construct confidence intervals, which are generally always reported for ATE estimation?  Can one plug in quantities for equation 12 from in-sample estimates?  Can the authors compare widths and coverage properties of confidence intervals constructed in this way?
*    The authors are noncommittal about regularizing orthogonality of both $Y(0)$ and $Y(1)$: appendix A indicates it’s not necessary if plugging the true outcome and propensity score model: however, in practice, does it make a difference?
*    A small complaint about terminology: the authors state in 3.1 that “the inner product in (5) is the empirical covariance between $Y(t)$ and $T$ given $X$.” However, equation (5) is not a function of X so this statement cannot exactly hold.  The orthogonality constraint requires some removal of conditioning, which the authors should address.

---

> ### Author Response · Authors · 2020-11-13
> **RE: A novel method that performs admirably; some clarification needed in places**
>
> We highly appreciate your valuable comments. Below, we respond to your questions.
>
> 0) Ablation analysis DONUT vs. TARNet: As correctly mentioned, TARNet is DONUT without the $\Omega_{OR}$ regularizer. As such, we can in general interpret the results DONUT vs. TARNet as an ablation analysis on the $\Omega_{OR}$ regularizer. Similar to previous work, we took the numbers for TARNet in Table 1 for the datasets HIDP, TWINS, and Jobs from Shalit et al. While this might not match all experimental variables, it ensures comparisons against state-of-the-art performance levels. To provide an ablation analysis, we also ran TARNet (with all experimental variables excatly matched to those of DONUT) on the ACIC collection of almost 100 datasets (see Table 2). Here we find the following: We observe on ACIC that DONUT performs superior to TARNet (DONUT without regularizer and identical experimental variables) across a wide range of datasets.
>
> 1) We do not move from (31) to (32). We move from (31) to $\hat{\psi} = \hat{\mathbb{P}}(m(Z; \hat{\eta}))$, where $m(Z; \hat{\eta})$ is given by (32).  To elaborate more:
> Equation (31) gives an analytical expression of our estimator for the average treatment effect, i.e.,
> \begin{equation}
> \hat\psi = \frac{\frac{1}{n}\sum_{i=1}^n (Y_i - \hat f(X_i, 0))(T_i - \hat\pi(X_i))}{\frac{1}{n}\sum_i^n T_i(T_i-\hat\pi(X_i))}
> \end{equation}
> This is identical to the empirical expectation of the random variable $m(Z; \hat{\eta})$, which is given in (32). This means $\hat{\psi} = \hat{\mathbb{P}}(m(Z; \hat{\eta}))$, and
> $m(Z; \hat{\eta})$ is given by:
> \begin{equation}
> m(Z; \eta) = \frac{(Y - f(X, 0))(T-\pi(X))}{\mathbb{P}(T(T-\pi(X)))},
> \end{equation}
> where $\eta=(\pi, f)$ denotes the nuisance functions. Hence, we do not claim that (31) and (32) are identical.
> 2) $f^\epsilon$ is a perturbation of $f$ as defined in (10). However, under correct model sepcification, the expectation of $f^\epsilon$ equals $f$. This can be seen by taking the
> expectation of $f^\epsilon$ in (10).
>
> 3) Table 2 present the results for estimating average treatment effects on the ACIC 2018 datasets. On this collection of 97 datasets, we compare
> Dragonnet (the current state-of-the-art model), TARNet (DONUT, but without the regularizer $\Omega_{OR}$), and DONUT. The reasons for including these experiments is
> twofold: (i) we compare DONUT to the current state-of-the-art, Dragonnet, and observe that the state-of-the-art is outperformed by DONUT over a wide range of datasets.
> (ii) We compare DONUT to TARNet (DONUT without regularizer $\Omega_{OR}$) as an ablation analysis (see 0) above). We observe that the regularizer $\Omega_{OR}$ substantially improves
> the estimation of the average treatment effect.
>
> 4) We do not assume that the treatment effect is deterministic given $X$. In the description of the pseudo outcome, the identities are meant "in expectation". We will clarify this in the revised version of the manuscript.
> The assumption of a constant treatment effect is not at play in the entire paper. It is only assumed to prove asymptotic normality in Theorem 1. This assumption can be easily relaxed to any specification of $\psi$ as long as it has finitely many parameters and given that the appropriate identification criteria hold. See Appendix D for details.
> We discussed the advantage of this assumption in the paper on page 5. The advantage of a constant treatment effect is that it explains why the asymptotic variance of $\hat{\psi}$ is smaller compared to other estimators. In particular, the difference in asymptotic variance to some estimators (e.g., the inverse probability weighted estimator) can be sizable if the propensity score is close to zero or one. Often, this difference is not offset by weaker restrictions imposed by heterogeneous treatment effects (see e.g., Vansteelandt & Joffe, 2014).
>
> 5) The construction of confidence intervals for our estimator could be addressed by using (12) and plugging in estimators for the nuisance functions. This could lead to (comparably) small confidence intervals, especially as our estimator offers small asymptotic variance. However, this was not the objective of our paper.
>
> 6) As indicated in Appendix A, it is sufficient to regularize one of the potential outcomes, since regularizing one potential outcome, ensures that the other potential outcome is regularized as well. We ran experiments to see whether regularizing both potential outcomes makes a difference. We made the following observation: In general, regularizing both potential outcomes does not improve the performance. But, on smaller datasets like IHDP, the increased number of parameters that have to be estimated decreases the performance. For this reasons, we regularize one potential outcome in our paper.

---

> > ### Author Response · Authors · 2020-11-13
> > **RE: A novel method that performs admirably; some clarification needed in places**
> >
> > 7) Removing the conditioning on $X$ (i.e., integrating over $X$) does not change the orthogonality constraint in (5). This is the case, since (due to unconfoundedness) the conditional covariance has to be zero, and integrating over quantities that are zero yields zero again. We will clarify this in the revised version of the manuscript.

---

### Decision · Program_Chairs · 2021-01-07
**Final Decision**

**Decision:**

Reject

**Comment:**

This paper proposes a regularization term that enforces the orthogonality between (i) a residual between the observed outcome and its estimator and (ii) the treatment and propensity score. The method empirically performs competitively. However, there seems to exist a gap between the proposed method and the assumptions made to provide theoretical guarantees (e.g., R3, R2). R4 was also concerned about the issue and adjusted his/her score accordingly. Even though the authors provide a detailed discussion on most of the reviewers' concerns, some of the problems remain unresolved. Further, unlike other papers submitted to ICLR, the authors did not actually update the paper such that we could check whether the revisions were adequately made. As such, I believe this paper is not quite ready for publication in its current form.

---

> ### Author Response · Authors · 2021-01-18
> **Re: Final Decision**
>
> We highly appreciate the overall positive feedback and that the reviewers found our idea very interesting. We carefully took into account any points raised by the reviewers and, based on it, revise our paper. In particular, we changed the following in our revised version of the paper:
>
> 1. We extended the discussion on related work. In particular, we incorporated more discussion on methods from the statistics literature for estimating average treatment effects and how they are adapted in the machine learning literature to estimate the average treatment effect.
>
> 2. We extended the current baselines with the above mentioned baselines from the statistics literature to give a more detailed view on the benefits of our method. Our method remains the best performing method across all datasets.
>
> 3. In the Section 4.2, where we explained our regularization framework, we clearly stated how the untreated outcome can be written with the treatment effect (without involving an expectation) and incorporated this into the regularization framework.
>
> 4. For better understanding of the method, we added a subsection (Section 4.3 in the revised manuscript) , where we give an intuition on why orthogonal regularization improves the estimation of the ATE.
>
> 5. We explained the limits of the theoretical result in Theorem 1, but clearly undermined that our method does not rely on the assumptions made in Theorem 1. We clearly note that our method work also outside of the (potentially) restrictive assumptions of Theorem 1, which we show in Section 5 in our experiments. We stated the theoretical result for completeness of our work.
>
> 6. In order to clearly  show the benefit of orthogonal regularization, we ran an ablation study comparing the estimation performance of DONUT (with the orthogonal regularizer  $\Omega_{\text{OR}}$) and DONUT without the orthogonal regularizer $\Omega_{\text{OR}}$. All other experimental variables are fixed. We ablation study is run on the same datasets as the comparison with the baselines: IHDP, Twins, and Jobs. The ablation study shows that orthogonal regularization improves estimation of the ATE substantially across all datasets.
>
> 7. We added a sensitivity analysis on the hyperparameter $\lambda$ to further investigate the robustness of our results. We observe that the results are very robust with regards to the hyperparameter $\lambda$.
>
> We thanks the reviewers for their valuable feedback, which allowed us to improve our work.